# Machine learning-enabled constrained multi-objective design of architected materials

Bo Peng [1,2,7], Ye Wei [3,7,8] ✉, Yu Qin [4,7,8] ✉, Jiabao Dai[1,2], Yue Li [5], Aobo Liu[1,2], Yun Tian[6], Liuliu Han [5], Yufeng Zheng [4] & Peng Wen [1,2,8] ✉

Architected materials that consist of multiple subelements arranged in particular orders can demonstrate a much broader range of properties than their constituent materials. However, the rational design of these materials generally relies on experts' prior knowledge and requires painstaking effort. Here, we present a data-efficient method for the high-dimensional multi-property optimization of 3D-printed architected materials utilizing a machine learning (ML) cycle consisting of the finite element method (FEM) and 3D neural networks. Specifically, we apply our method to orthopedic implant design. Compared to uniform designs, our experience-free method designs microscale heterogeneous architectures with a biocompatible elastic modulus and higher strength. Furthermore, inspired by the knowledge learned from the neural networks, we develop machine-human synergy, adapting the ML-designed architecture to fix a macroscale, irregularly shaped animal bone defect. Such adaptation exhibits 20% higher experimental load-bearing capacity than the uniform design. Thus, our method provides a data-efficient paradigm for the fast and intelligent design of architected materials with tailored mechanical, physical, and chemical properties.

Architected materials are one of the most widely adopted engineering materials. Due to their excellent mechanical performance and adaptable properties, architected materials are very popular in many fields, such as those of lightweight structures[1–7], acoustics[8], battery electrodes[9], electromagnetics[10–12], and tissue engineering[13–17]. Moreover, recent progress in 3D printing has further enabled the customized and inexpensive fabrication of complex material geometries. Despite the broad applicability and immense potential of architected materials, designing them is particularly difficult. The traditional method generally relies on numerical simulation, theoretical analysis, and topology optimization (TO). These rule-based undertakings are usually exhausting and time-consuming, and the performance of resultant designs highly depends on the designer's professional knowledge and their initial guesses[18–21]. Recently, machine learning

(ML) has emerged as a promising technique to circumvent this problem and find the optimal solution without any prior knowledge requirements[22–27]. Furthermore, active learning that combines machine learning and simulations or experiments to tackle optimization problems is an emerging topic at the frontier of science[28]. It introduces an iterative ML algorithm that identifies high-value solutions with fewer labeled data[29,30]. Such efficiency is highly desirable when the data is sparsely distributed. However, some of these methods mainly focus on 2D-structure-related problems, while others use Bayesian optimization to solve low-dimensional problems or focus on an unconstrained single objective[31–34]. The efforts toward solving high-dimensional multi-objective problems are often obfuscated by the data sparsity, the enormity of the search space, and stringent external constraints.

[1]State Key Laboratory of Tribology in Advanced Equipment, Tsinghua University, Beijing, China. [2]Department of Mechanical Engineering, Tsinghua University, Beijing, China. [3]Institute for Interdisciplinary Information Science, Tsinghua University, Beijing, China. [4]Department of Materials Science and Engineering, Peking University, Beijing, China. [5]Max-Planck-Institut für Eisenforschung, Düsseldorf, Germany. [6]Department of Orthopaedics, Peking University Third Hospital, Beijing, China. [7]These authors contributed equally: Bo Peng, Ye Wei, Yu Qin. [8]These authors jointly supervised this work: Ye Wei, Yu Qin, Peng Wen. ✉ e-mail: ye.wei@rwth-aachen.de; qinyu95@126.com; wenpeng@tsinghua.edu.cn

In this work, we introduce an active learning route that effectively combines a generative model with physical simulation to perform a high-dimensional multi-objective optimization under various constraints (Supplementary Fig. 1), commonly encountered in many real-world engineering designs[35]. As demonstrated in Fig. 1, our approach consists of three main parts: (1) generative architecture design (GAD). In this step, GAD leverages the encoder-decoder neural network (autoencoder) to generate architecture sets with unknown properties. The autoencoder learns an effective representation of the high-dimensional data in an unsupervised manner, which converts the exploration in a high-dimensional design space into a lower one. This method has been proven to be a revolutionary technique in materials discovery[36,37]. (2) Multi-objective active learning loop (MALL). MALL evaluates the generated dataset and searches for high-performance architecture by recursively querying the finite element method (FEM). (3) 3D printing and testing. Finally, we fabricate the ML-designed architected materials via a specialized 3D printing technique (laser powder bed fusion) and experimentally verify the corresponding mechanical properties. We call the overall method "GAD-MALL". The technical details are described in the "Methods".

## Results

### Multi-objective active learning algorithm

We applied the GAD-MALL approach to a multi-property optimization problem with clinical importance—bone grafting implants. Bone is a typical architected material primarily consisting of cortical and cancellous parts, with the elastic modulus ($E$) ranging from 0.03 to 30 GPa depending on the bone mineral density and varying according to age, sex, and race[38]. Although bone can repair itself, a bone defect of a critical size necessitates a grafting implant to support the load and induce bone growth. Metals are the promising choice for bone implant materials due to their excellent mechanical properties. However, the $E$ of the existing metal bulk materials is much greater than that of the bones (i.e., titanium −100 GPa; iron −200 GPa, etc.), which results in the stress shielding effect and impedes the recovery of the bone[39]. One effective solution is introducing a 3D-printed scaffold architecture to lower the $E$. The geometrical shape and mechanical properties of the scaffold should be comparable to those of the individual defective bone to provide reliable structural support and smooth stress conduction. The mechanical response of the scaffold can be represented by a compressive stress-strain curve. The slope of the linear section of the curve represents $E$, which measures a material's ability to resist external stress before being deformed permanently, and the yield point with 0.2% strain represents the yield strength ($Y$), which quantifies the maximum resistance before the onset of nonreversible deformation. Therefore, we need to optimize several targets under external constraints: (1) the $E$ of the scaffold implant must match that of the bone. (2) the $Y$ must be as high as possible to sustain the bone. (3) the resultant structures must be biocompatible and 3D-printable. An important prerequisite is that the simulated properties agree with experimental observation within an acceptable error range. Therefore, several replicates (to ensure reproducibility) of candidate materials were manufactured and tested, by which the experimental measurements ($E$ and $Y$) were used to calibrate the FEM parameters such that the error between the simulation and experimental results was less than 10% (see "Methods"). In addition, the overall weight of the scaffold should not go beyond a certain threshold since a minimum usage is always required considering long-term biosafety.

Moreover, to balance complexity and computing efficiency, we adopted the $3 \times 3 \times 3$ cubic arrangement of the gyroid units as the model input for the optimization task (see Methods for the structure generation). Therefore, this problem can be treated as a high-dimensional multi-objective optimization problem, presenting an exponential difficulty due to the "curse of dimensionality"[42]. The gyroid geometry is categorized in the triply periodic minimal surfaces (TPMS) family—it is an ideal porous structure for bone scaffolds due to its high interconnectivity, smooth surface, and mathematically adjustable geometrical attributes[40,41]. The multi-objective topology optimization can also achieve multiple mechanical targets by optimizing the topology under constraints, but such change often compromises other desirable biological functions and leads to overly complicated designs with limited printability or suboptimal mechanical performance[42–44]. The gyroid unit is carefully chosen since it has many advantages for bone implants, e.g., it has a Gaussian curvature close to 0 and a saddle shape similar to that of bone trabeculae, which has been shown in related studies to better promote tissue growth compared to other structures[45]. In addition, various works have pointed out that the distribution of TPMS units can influence the mechanical properties, but so far none has quantitatively evaluated such correlation, not to mention the optimization[46]. Hence, to guarantee biocompatibility, 3D-printability, and service applicability, we adjust the size of the gyroid unit (porosity) within the scaffold instead of changing the topology of the subunit, resulting in a geometrical alteration that modulates the overall mechanical properties.

### Applications to orthopedic implants

The properties of the architected materials are determined by both the scaffold architecture and the constituent materials. Ti6Al4V (Ti) and pure zinc (Zn) were used as constituent materials for orthopedic implants. Ti alloy is bioinert in human bodies and has been the de facto

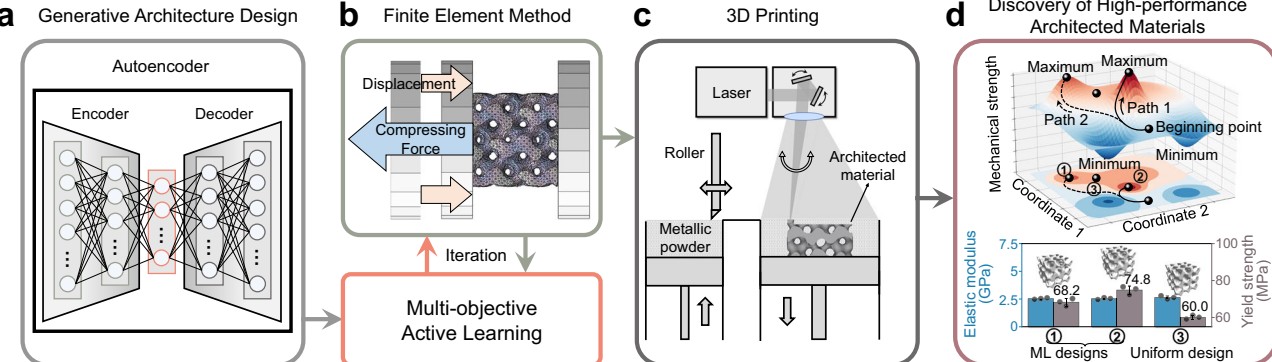

**Fig. 1 | An overview of the proposed workflow (GAD-MALL, i.e., Generative architecture design−multi-objective active learning Loop). a** The neural network proposes candidates with unknown properties. **b** The machine-learning (ML) algorithm interactively queries the finite element methods (FEM) to propose new designs. **c** The 3D printing technique fabricates the proposed architectural design. **d** GAD-MALL explores the design landscape of architected materials and discovers various high-performance architected materials (mean ± SD, $n = 3$). Source data are provided as a Source Data file.

choice for 3D-printed orthopedic implants, achieving successful clinical application to repairing bone defects. Biodegradable Zn provides an alternative option to bioinert materials and is regarded as promising for addressing the clinical concerns associated with permanent existence and secondary surgery[47]. Such features are especially desirable for bone regeneration. As both materials are worthy of investigation, to demonstrate the effectiveness and general applicability of the GAD-MALL framework, we designed two optimization tasks for both constituent materials and applied the learned design principle to the real bone replacement architecture. Specifically, the Ti alloy scaffolds were assigned a high $E$, while the pure Zn scaffolds had a low $E$, indicating different clinical needs based on the constituent materials. In addition, two tasks were given different initial data distributions to demonstrate that GAD-MALL can work under different initial conditions. Notably, all tasks were completed in one week with the current hardware setup, as tasks in the clinical scene are usually time-constrained. In the following section, we begin with the Ti cubic scaffolds.

## A data-efficient route toward high-performance structure

To mimic the mechanical behavior of trabecular and compact bones, the task was to design high-$Y$ scaffolds with $E = 2500$ MPa and 5000 MPa (E2500 and E5000). The uniform-designed scaffolds at $E = 2500$ MPa and 5000 MPa set the "gold criteria" for the mechanical performance of the scaffolds (see Supplementary Fig. 2 for the design protocol of a uniform scaffold with specific $E$), the expert and uniform designs refer to two designs: one is a scaffold generated using topological optimization (see Supplementary Fig. 3) and the other uniformly sized Gyroid subunits which has been adopted in other studies[44,48,49]. GAD-MALL stopped if the $Y$ of the designed scaffold significantly surpassed the "golden criteria" (termed the "treasure" scaffold) or the learning process showed no further progress. The initially labeled dataset was composed of merely 95 data points (the simulation took ca. 7 days, hardware specified in the Methods section), since the predictive model based on this dataset already showed good performance on the testing dataset. Figure 2a demonstrates the good performance of the 3D convolutional neural networks (3D-CNNs) on the test dataset (uniformly sampled from the labeled dataset) in the 1st round and last round, in which both 3D-CNNs demonstrate high accuracy (coefficient of determination, i.e., $R^2 > 0.92$). A more detailed performance evaluation of the models used in this study can be found in Supplementary Fig. 4 (Performance evaluation of the 3D-CNNs on the Ti test dataset), Supplementary Fig. 5 (performance evaluation of the 3D-CNNs on the Zn testing dataset), Supplementary Fig. 6 (Training of the 3D convolutional autoencoder), and Supplementary Fig. 7 (the average negative log-likelihood versus the number of clusters in the Gaussian mixture model). Figure 2b shows that the scaffolds had been precisely manufactured—the cross-sections of the micro-computed tomography (Micro-CT) of the scaffolds largely overlapped with that of the designs (more Micro-CT can be find in Supplementary Fig. 8). The superior efficiency of GAD-MALL is evidenced by comparison with other baseline methods under current problem settings as well as under a toy problem setting, in which the global optimal solution is known (Fig. 2c, Supplementary Methods, and Supplementary Fig. 9). In the current design problems, the GAD-MALL iteration-strength curve demonstrated a clear upward trend of mechanical strength improvement, while both curves of the random-search-based and Bayesian-optimization-based active learning were relatively flat, showing no notable improvements.

Figure 2e shows the overall data distribution in terms of $E$ and $Y$ with the treasure scaffolds indicated by blue stars. Each active learning iteration is characterized by colored ellipses. Figure 2d, f demonstrate two distinct exploration paths for two different tasks (the detailed results of each learning round are described in Supplementary Fig. 10

and Supplementary Table 1). The E2500 exploration path shows a steady upward trend, and GAD-MALL quickly discovered the treasure scaffolds at the 3rd and 5th rounds with more than a 30% increase in $Y$. However, the E5000 task was more complicated—the learning process exhibited a downward trend before it recovered and found the treasure scaffolds. Specifically, the batches from the 1st to 3rd rounds either fell out of the target $E$ region or had inferior $Y$-values. The 4th-round batch finally hit the target of $E$, albeit $Y$ was not notably better than that of the uniform designs. Finally, the treasure scaffolds were discovered on the 5th and 6th rounds. This oscillatory trend is likely due to the sparsity of data within this range (with only two initial data points available).

The experiments confirmed the discovery—the ML-designed scaffolds (A1–A4) showed better performance than the uniform-designed scaffolds (H1 and H2). More details are available in Fig. 2g, Supplementary Fig. 11, and Supplementary Table 2. The experimental strain–stress curves of the A1 and H1 scaffolds are also displayed in the inset in Fig. 2g. To understand the ML design, we further analyzed the ML-designed scaffold by extracting the corresponding regression activation map (RAM) and performing FEM mechanical analysis. As an illustrative example, we applied the RAM to the $Y$-predicting 3D-CNN model to reveal the driving mechanism behind the high $Y$ of the A1 scaffold. RAM is a variant of a classification activation map that extracts the last convolutional layer to visualize the discriminative regions used by a 3D-CNN to predict the output[50]. In this case, the RAM highlights the scaffold's spatial characteristics that correlate to its mechanical strength, identifying the regions that contribute to the enhancement of strength. Figure 2h demonstrates the A1 scaffold geometrical structure, the corresponding porosity matrix, and the RAM (see Supplementary Fig. 12 for more results). The RAM implies that the "attention" distribution extracted from the 3D-CNN resembled a heterogeneous "face-centered" lattice. Indeed, a closer look at the A1 scaffold revealed that the gyroid units at each face center of the scaffold showed a minimal porosity (0.3). This observation indicates that instead of uniformity, a heterogeneous scaffold with more materials distributed at the face centers could significantly enhance the strength (see Supplementary Fig. 13 for more results). Moreover, from a macroscopic point of view, the strength of a typical porous structure can be approximated by the Gibson-Ashby equation[51]:

$$Y = C(1 - p)^{\alpha} Y_0 \tag{1}$$

where $Y_0$ stands for the strength of the constituent material, $C$ represents a geometry-related parameter, $p$ is the porosity of the unit, and the exponent $\alpha$ relates to the deformation behavior of the structure. According to the FEM calculated data, we fitted the curve of strength $Y$ as a function of $p$ for ML and uniform-designed scaffolds and found $a_{ML} = 2.11$, $a_{UD} = 1.86$, $C_{ML} = 0.84$, and $C_{UD} = 0.64$, in which UD stands for uniform design.

The ML-designed scaffold had a larger $a$ and $C$ than the uniform design. Generally, increasing the mechanical anisotropy of a porous structure leads to an increase in the exponential factor $a$, while an increase in parameter $C$ can be found in the material distribution in favor of the load direction[52].

Microscopically, FEM analysis confirmed the above observation. Figure 2i shows the distribution of von Mises stress and hydrostatic pressure of the A1 and H1 scaffolds (see Supplementary Fig. 14 for more results). Compared with the H1 scaffold, the A1 scaffold endures a much weaker effect of stress concentration; moreover, more struts of the A1 scaffolds are compressed rather than stretched. The ML model preferentially places more materials on the face center of the scaffolds, which optimizes the stress distribution and improves the structural strength with increasing limited mass. Hence, GAD-MALL was able to find the optimal architectures by efficiently learning from a few initial data points.

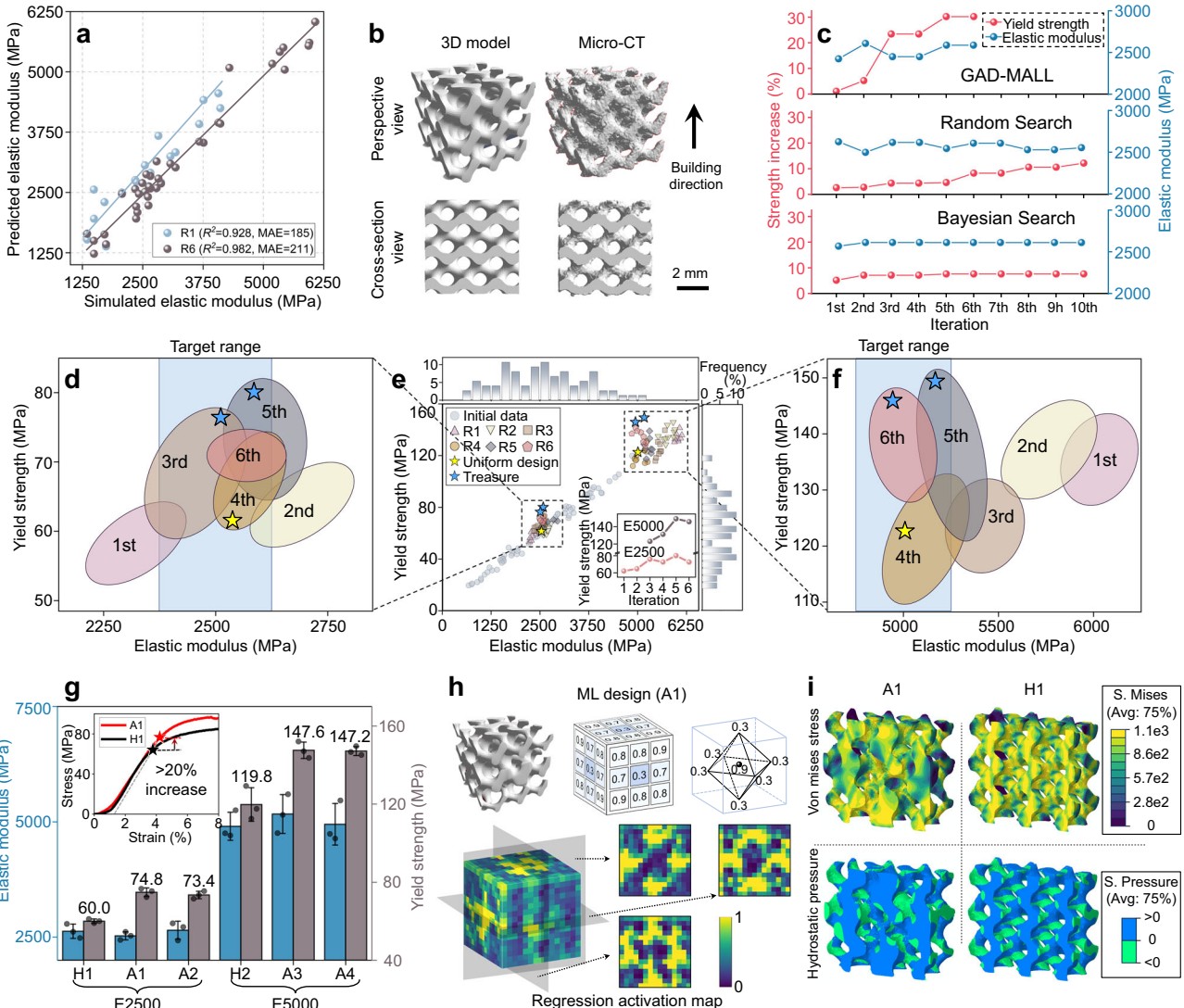

**Fig. 2 | Data-efficient learning of high-performance scaffolds. a** The regression plots (1st and last rounds of active learning) of the 3D convolutional neural networks (3D-CNNs) for elastic modulus ($E$). Both 3D-CNNs demonstrate excellent accuracy on the testing set, showing a low mean absolute error (MAE) and a high coefficient of determination ($R^2$). **b** Micro-computed tomography (Micro-CT) shows that the designated scaffolds were accurately manufactured. **c** Baseline comparison of GAD-MALL (Generative Architecture Design−Multi-objective Active Learning Loop) with random-search and Bayesian optimization. **d, e, f** The overall finite element method (FEM) simulation data distribution in terms of the $E$-yield strength ($Y$) plot. The inset of **e** shows the maximal $Y$-values corresponding to structures designed in each iterative round while concurrently meeting the pre-defined target $E$ range. Initial data points are marked as light blue dots. Six rounds of active learning data points are represented by various symbols (R1−R6). The uniform design and GAD-MALL treasure points are marked as golden and blue stars, respectively. The colored ellipses represent the area covered by 6 rounds of active learning data. The blue shaded area indicates the target range of $E$ (i.e., target

$E$-value ± 5%). **g** Comparison of the experimental $E$ and $Y$ between machine-learning (ML)-designed (A1, A2 for E2500 and A3, A4 for E5000) and uniform-designed (H1 for E2500, H2 for E5000) scaffolds. The inset shows the experimental strain–stress curves of the A1 and H1 scaffolds. The dashed line within the inset is derived by translating the linear segment of the stress-stress curve horizontally by a 0.2% strain offset and is used to obtain the yield points of the curves. The $Y$ of the ML-designed scaffolds was obviously higher than that of the uniform designs. Data are presented as mean values ± SD, $n$ = 3. **h** The upper figures show the mathematical model of the A1 scaffold and its porosity matrix. The lower figures contain the 3D view and three cross-section views of the regression activation map (RAM). The RAM reveals a "face-centered" lattice in the A1 scaffold, implying its prominent role in enhancing $Y$. This face-centered lattice is displayed in the upper right part of the figure. **i** Numerical compression analysis. Here, we show the $y$–$z$ cross-sections of A1 and H1 scaffolds in terms of von Mises stress and hydrostatic pressure under 10% deformation. Source data are provided as a Source Data file.

## Learning without prior data in the target range

To demonstrate the robustness of the GAD-MALL approach, we designed a learning task by which GAD-MALL found the appropriate scaffolds "from scratch"−the initial Zn dataset did not contain any prior data points in the target range by design. The task of this section was to design high-$Y$ scaffolds at $E$ = 500 MPa and 1000 MPa (E500 and E1000), targeting the replacement of cancellous bone. Again, the uniform-designed scaffolds at $E$ = 500 MPa and 1000 MPa set the "golden criteria".

Figure 3a–c illustrate the $E$−$Y$ distribution of the initial data (marked as light blue dots) and the results from each active learning round characterized by colored ellipses (the detailed results of each learning round are described in Supplementary Fig. 15 and Supplementary Table 3). Figure 3b demonstrates that the GAD-MALL exploration paths of the missing data were complicated, exhibiting back-and-forth trends. For the E500 task, the $E$ distribution of the 1st round shows a significant standard deviation. It is noteworthy that some scaffolds from the 1st round had already reached the target

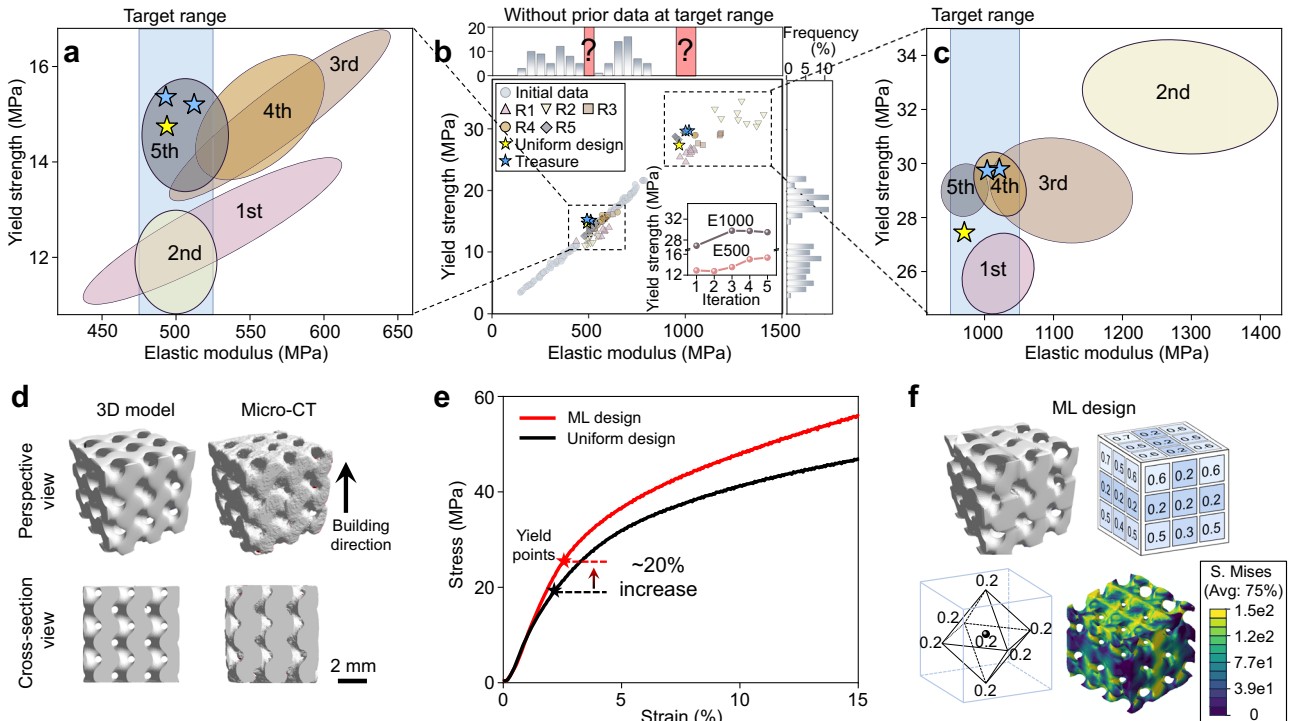

**Fig. 3 | Learning without prior data at the target range. a, b, c** The elastic modulus (*E*)−yield strength (*Y*) distribution of finite element method (FEM) simulation results. The inset of **b** shows the maximal *Y*-values corresponding to structures designed in each iterative round while concurrently meeting the predefined target *E* range. Initial data points are marked as light blue dots. Five rounds of active learning data points are represented by various symbols (R1–R5). The uniform design and GAD-MALL treasure points are marked as golden and blue stars, respectively. The colored ellipses represent the area covered by 5 rounds of active learning data. The blue shaded area indicates the target range of *E* (i.e., target *E*-value ± 5%). **d** Micro-computed tomography (Micro-CT) shows that the designated Zn scaffolds were manufactured with good precision. **e** The experimental strain–stress curves of the machine learning (ML) and uniform-designed scaffolds. The ML design yielded a 20% increase in *Y*. **f** The porosity of the ML-designed scaffold reached the lower limit (0.2) at the face centers and the center of the scaffold. Similar to the ML-designed Ti scaffold, the compression analysis shows that the low-porosity units of the ML-designed Zn scaffold had lower stress concentrations. Source data are provided as a Source Data file.

E ≈ 500 MPa. The 2nd round shows improvement—the overall standard deviation was significantly reduced (from 52 to 19 MPa). While all scaffolds' *E* values were located at approximately 500 MPa, the *Y* values were still 30% less than the gold criteria. In the following rounds, the exploration path reached a plateau, and the selected candidates were slightly better than the golden criteria (14.8 MPa). The E500 task was terminated after the 5th round since no further progress was observed (see inset of Fig. 3b).

On the other hand, GAD-MALL excelled at the E1000 tasks, outperforming the golden criteria by a large margin. More specifically, the 1st round already showed promising results, in which all scaffolds exhibited the targeted *E*, although with slightly worse *Y* (≈10%). The subsequent round witnessed a significant decrease in porosity (Supplementary Table 3), which in turn remarkably enhanced *Y*. However, the reduced porosity resulted in another problem—the *E* increased to 1200 ~ 1400 MPa. GAD-MALL incorporated this knowledge into the database in the subsequent learning process.

Eventually, the average porosity increased, and the treasure scaffolds were discovered in the 3rd and 4th rounds. The entire learning process took approximately 9 days, and the mechanical properties of the resultant designs are tabulated in Supplementary Table 4.

Figure 3d illustrates the model and Micro-CT of an exemplary ML-designed scaffold (see Supplementary Fig. 16 for more results). From the cross-section view, the model and manufactured sample were shown to agree with each other. The ML-designed scaffolds were manufactured, and their mechanical properties were measured experimentally (Fig. 3e and Supplementary Fig. 17). The ML design had a significant performance advantage over the uniform design, whose *Y*

(26.4 ± 0.7 MPa) exceeded the golden criteria (21.7 ± 1.8 MPa) by a large margin of 21.6%, with a slightly lower porosity. As the *E* and *Y* of the bulk Zn was less than those of the Ti alloy, the Zn scaffold still had a lower porosity even though the target *E* was only 1000 MPa. Similar to the Ti scaffold, the FEM analysis in Fig. 3f shows that the low-porosity face-centered units in the ML-designed scaffold had less stress concentration, leading to enhanced strength (see Supplementary Fig. 18 for more results). Since the face-centered and the central unit of the Zn scaffold had reached the lower limit (porosity = 0.2) and the excess weight was allocated to the central and the ridge-center unit of the cubic scaffold, the *E* of the scaffold did not hit the targeted *E* range (*E* = 1000 ± 50 MPa). Thus, the central and ridge-center units promoted *E* to the target range without decreasing *Y*. In this task, we showcased that GAD-MALL was able to find the optimal architecture even when the initial data distribution and the constituent material were considerably different from those in the previous section. Such robustness is highly desirable since clinical situations can be quite variable—the patient data (target material and mechanical range) are often unknown beforehand, and the initial data can have various distributions.

## ML-inspired anatomic bone implants

Most real-world bone implants require scaffolds in anatomical shapes that fit to the defective bone. Figure 4a, b shows a large, irregularly shaped bone defect in a New Zealand rabbit model animal model—a defect of critical size (30 mm) occurred in the middle part of the tibia. Figure 4c shows the 3D shape of the tibia, which was acquired through Micro-CT scanning. It is difficult and time-consuming to find the optimal scaffold architecture to fit the shape, whether by experimental

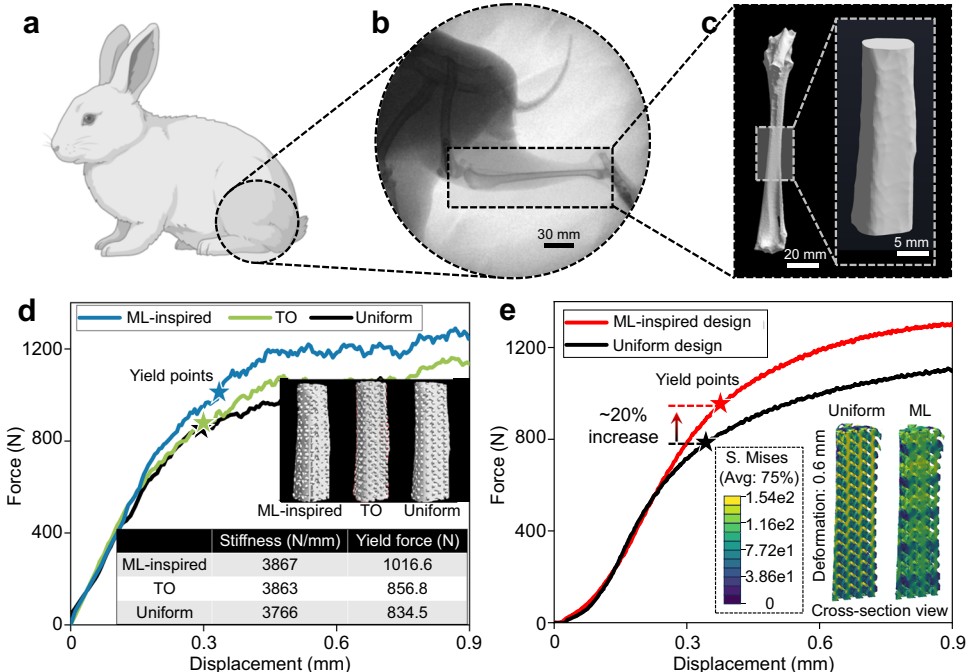

**Fig. 4 | Anatomic bone fixation with machine learning (ML)-inspired design.** **a**, **b** A 30 mm bone defect in a New Zealand rabbit's middle part of the tibia. **c** Micro-computed tomography (Micro-CT) of the tibia. **d** Finite element method (FEM) simulated displacement-force curves of ML-inspired, topology optimization (TO), and uniform designs. **e** Experimental displacement-force curves of the ML-inspired design versus uniform design. The inset shows the cross-sections of von Mises stress under 0.6 mm deformation for both designs. The cartoon in Fig. 4a was created with BioRender.com. Source data are provided as a Source Data file.

or numerical trials since there are many possible choices. Here, we demonstrate how a machine-learned design principle can be readily adapted to a clinical scene through a facile machine-human design workflow.

Concretely, to use the ML-designed cubic scaffold for a larger implant for large, irregularly shaped bone defect fixation, our workflow constituted the following two steps (Supplementary Fig. 19): (1) Using the ML-designed cubic scaffold as the basic unit, we manually created a cuboid of $3 \times 3 \times 9$ units with width, length, and height of 18 mm, 18 mm, and 54 mm, respectively. (2) Subsequently, we caved out an irregularly shaped scaffold from the interior of the cuboid that matched the bone shape. Then we performed a FEM study of TO (Supplementary Fig. 3), ML, and uniform designs to assess their mechanical performance, and the simulated results show that the load-bearing capacity of the ML design is considerably higher than that of the other two designs (Fig. 4d). The experimentally validated mechanical behaviors at the macroscale could be characterized by the displacement-force curves in Fig. 4e and Supplementary Fig. 20, which confirmed that the stiffness of uniform-designed and ML-inspired implants was almost the same, while the ML-inspired implant's load-bearing capacity (indicated by stars) was considerably higher (20%). The von Mises stress distribution, given in the inset of Fig. 4e, showed that the overall stress (under 0.6 mm deformation) of the ML-inspired design was considerably higher than that of the uniform design (see Supplementary Fig. 21 for more results). With the same bone shape and deformation, the higher inner stress accumulation of the ML-inspired design indicated stronger support of the bone implant. Therefore, the strengthening effect of the ML-designed face-centered lattice was accumulative; a large structure made up of many individual strengthened cubes still demonstrated better load-bearing capacity than the uniform design of the same scale.

## Discussion

This work demonstrates a multi-objective active learning approach for designing 3D-printed architected materials with generative models and 3D neural networks under several external constraints. With only 95 initial fine-tuned FEM simulation data points, our approach quickly discovered high-performance architected materials. Thus, by fusing high-precision simulation, ML, and 3D printing, our framework was developed into a powerful and robust tool that excels at complex multi-objective architecture optimization. It represents a data-efficient, intelligent method that requires no prior knowledge and can be readily adopted in wide-ranging architected materials applications. While GAD-MALL can learn the structures described by a set of parameters and is capable of discovering the corresponding parameter-property relation, the structures without a clear mathematical description are beyond the capacity of this method. One could overcome such limitations by (1) introducing more architectural degrees of freedom in terms of parameters, such as taking more non-linear terms as an input variable (i.e., gradients, periodicity, etc.), or (2) using raw input representation such as 3D point clouds and voxelization, both of which could lead to the discovery of new families of metamaterials but requires more carefully design of the generative model and significant computational effort. Meanwhile, the application of other advanced generative models (such as vector-quantized variational autoencoder) warrants further investigation, as some of them show superior performance in generating high-quality samples. In the present case, the input representation and underlying data distribution of the unlabeled dataset are simple enough that the 3D convolutional autoencoder (3D-CAE) showed the best performance. In the future, advanced models as such hold great promise in learning more complex data with various underlying latent distributions, leading to the discovery of new metamaterials. Finally, we developed a synergistic machine-human design methodology that uses machine-learned small-scale, regular structures as subunits to create large-scale, irregularly shaped architectures. Overall, we anticipate that our methodology can be used for quickly designing architected materials where optimal responses to various stimuli are desirable, including mechanical, thermal, and chemical conditions or application requirements.

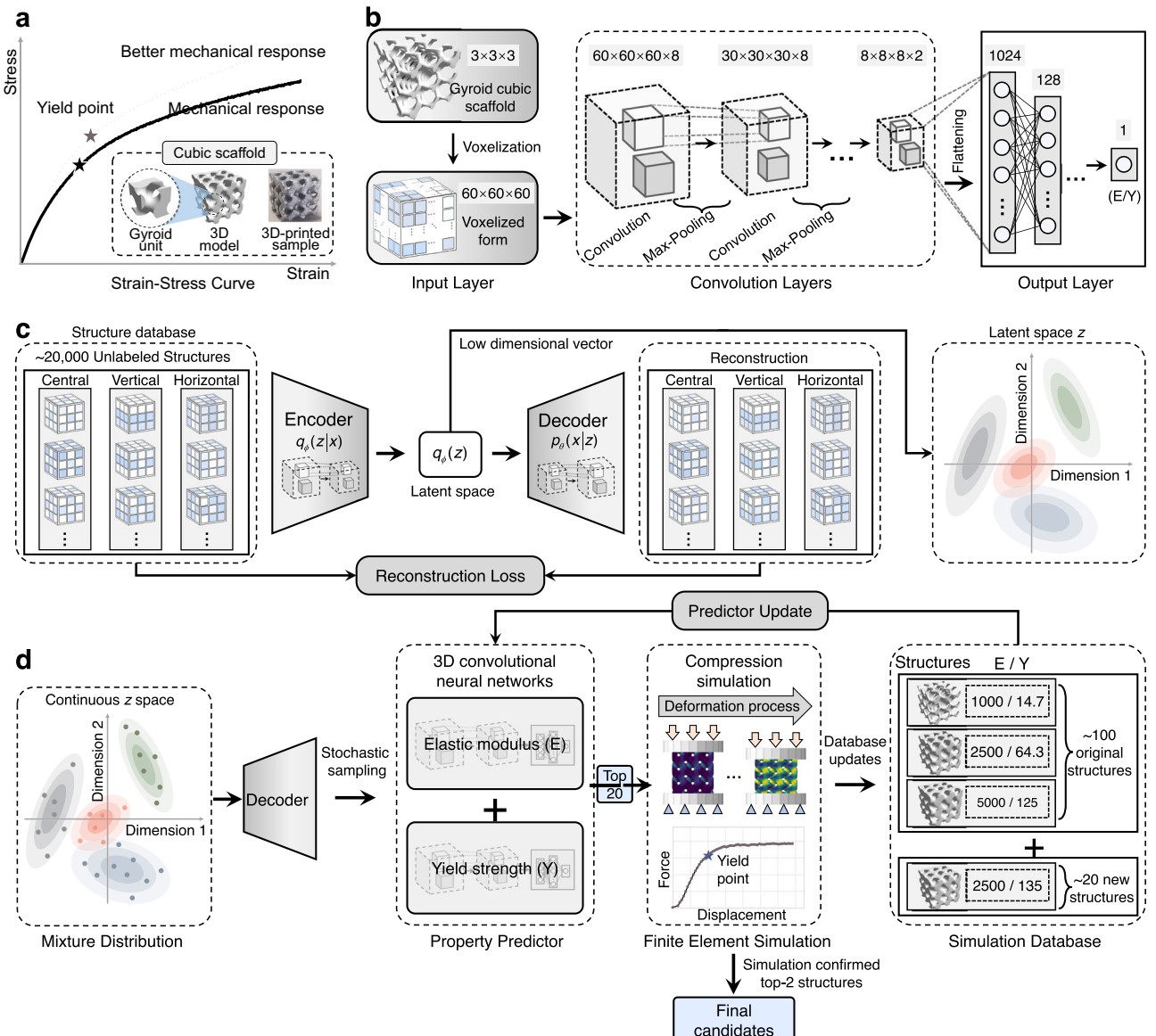

**Fig. 5 | The workflow of multi-objective active learning. a** The task is to design scaffolds with a better mechanical response—fixed elastic modulus ($E$) and maximized yield strength ($Y$). **b** The 3D convolutional neural networks (3D-CNNs) for predicting $E$ and $Y$. **c** The generative model for targeted scaffold generation. The encoder $q_\varphi(z\,|\,x)$ with parameters $\varphi$ took the scaffold porosity matrix as input, and the decoder $p_\theta(x\,|\,z)$ with parameters $\theta$ could act as a generator for proposing new scaffolds based on the learned latent $z$ representation. **d** Multi-objective active learning loop (MALL) for high-performance scaffold discovery. First, the sampling algorithm sampled new data points from the latent $z$ representation. Second, the decoder reconstructed the corresponding scaffolds so that the 3D-CNNs could infer their mechanical properties. Third, the most suitable candidates were selected based on the predicted $E$ and $Y$. Finally, the strain–stress curves of the selected scaffolds were calculated by the finite element method (FEM). New data were either fed back to the dataset or 3D-printed for further experiments.

## Methods

### Workflow of the GAD-MALL method

Figure 5b shows the models of the 3D-CNN for $E$ and $Y$ prediction. The 3D-CNN was designated for volumetric data representation learning[53,54]. It included three main components: input, convolution, and output layers. At the input layer, the scaffold structure was voxelized into $60 \times 60 \times 60$ pixels. A pixel can be in either the solid (1) or void (0) phase in the scaffold. The convolution layers consisted of a series of 3D convolution kernels that extracted high-level information about the scaffold, and the output layer provided the final prediction.

Figure 5c illustrates a 3D convolutional autoencoder (3D-CAE) with a typical two-neural network model, an encoder, and a decoder. Notably, a variational 3D-autoencoder with identical architecture showed much higher reconstruction loss for the same tasks, likely due

to the non-Gaussian underlying data distribution. Other input representations, such as voxelization, might serve the same purpose. However, it requires a sophisticated geometry optimization algorithm that removes the artifacts, without which the generated shapes might be able to satisfy the boundary condition or not even be 3D-printable. Therefore, GAD-MALL adopts a parameterized presentation, which guarantees the biocompatibility and printability of the generated shapes. We circumvented this problem by adopting the porosity matrix, a 3D matrix representation ($3 \times 3 \times 3$) that uniquely determines the overall geometry through gyroid equations. It measures the relative density (positive scalars) rather than the actual shape of the gyroid subunits, thereby allowing nonzero reconstruction errors. The encoder $q_\varphi(z\,|\,x)$ with parameters $\varphi$ compressed the porosity matrix into a hidden feature representation (8-dimension) using the neural encoder. Then, the decoder $q_\varphi(x\,|\,z)$ with parameters $\phi$ reconstructed the

output from the 8-dimensional hidden features. A lower dimension (e.g., 4-dimension) latent space was shown to suffer from high reconstruction error, while a higher dimension (e.g., 16-dimension) doubled the search space without a sufficient increase in reconstruction accuracy. Ultimately, 8 dimensions represented a balance between loss and efficiency (Supplementary Fig. 6).

Figure 5d shows the primary steps of the MALL workflow, which comprised three steps. First, scaffold generation was formulated as a process of sampling and reconstruction from the latent representation $z$. The sampling process required the latent representation to be modeled as a continuous probabilistic distribution. Second, the decoder $q_{\varphi}(x|z)$ reconstructed the porosity matrices from the sampled latent points, which were then converted to their original shapes in Cartesian space. The scaffold selection method was a variant of the epsilon-greedy search: in each sampling iteration, we sampled 2000 data points and selected those whose 3D-CNN-predicted $E$ met the target and whose 3D-CNN-predicted $Y$ exceeded the best data point in the current dataset, with a chance of epsilon (5%) that lower values were chosen. The selected data points would still be rejected if their weights were 15% higher than the preset criteria. Such a search method generally had a higher success rate than the Edisonian approach, which hinged on a trial-and-error search[55]. Last, the FEM calculated the $E$ and $Y$ of the queried scaffolds, and the results augmented the dataset, from which the 3D-CNNs were retrained for the following active learning round. The workflow stopped when all the preset criteria were met.

## TPMS structure generation

TPMSs and related structures are widespread in natural biological systems[56–63]. TPMSs are considered to be the ideal geometric shape to describe the biological form of the human skeleton[64]. Numerous studies have shown that the curved surfaces of TPMSs contribute to enhanced plasma membrane elongation during cell crawling and spreading[65–67]. In this study, we adopted the gyroid minimal surface structure, which is a member of the TMPS family. In addition to the abovementioned advantages of TMPSs, the helical surface structure of the gyroid unit makes the force distribution more uniform, leading to its excellent mechanical properties. The equation of the gyroid surface is as follows[68]:

$$\varphi_G \equiv \sin X \, \cos Y + \sin Y \, \cos Z + \sin Z \, \cos X = c \qquad (2)$$

The equation $\varphi(X, Y, Z)$ defines a surface evaluated at the isovalue (i.e., level-set constant) $c$ and has a topology similar to that of a minimal surface. $X = 2\alpha\pi x$, $Y = 2\beta\pi y$, $Z = 2\gamma\pi z$, $\alpha$, $\beta$, and $\gamma$ are constants related to the unit cell size in the $x$, $y$, and $z$ directions, respectively. In this work, we created the gyroid lattice based on the minimal surface by considering one of the volumes divided by the surface as the solid domain and the other as the void domain. This was done by considering the volume bounded by the minimal surface such that $\varphi(X, Y, Z) > c$ to create a solid-network lattice. The porosity of gyroid lattices can be graded by spatially varying the value of the level-set constant $c$ in Cartesian space depending on a certain function or tabulated data such that[69]:

$$\varphi_G > c(x,y,z) \qquad (3)$$

To achieve a smooth transition between units on edge (Supplementary Fig. 22), we describe the isovalue as a linear function along one of the Cartesian coordinates such that $c = Ax + B$, where $A$ and $B$ are constants. This smooth transition is a prerequisite for representing the actual geometric shape using a porosity matrix.

The scaffold contains 27 gyroid subunits in total, arranged as a $3 \times 3 \times 3$ cubic structure. The geometry of the scaffold is controlled by the $3 \times 3 \times 3$ porosity matrix. The porosity $c$ of each subunit can take discrete values from 20 to 90%, with an increment of 10%.

## Dataset generation

The unlabeled dataset consists of about 18,000 data points and is generated for the training of the 3D-CAE. In principle, the porosity of a subunit can take any value from 0 to 1. Therefore, the possible arrangement is infinite. To simplify the problem, we allow the scaffold's porosity to take discrete values from 20 to 90% with an interval of 10%. Nevertheless, there are still $7^{27}$ possible combinations in the design space. Three thousand matrices of various porosities are generated at each interval. For each interval, there are four kinds of arrangement in the database (Fig. 6a): central, vertical, horizontal, and random arrangements. There are also three types of porosity matrices: $2 \times 2 \times 2$, $3 \times 3 \times 3$, and $4 \times 4 \times 4$, which then all expand to a $12 \times 12 \times 12$ matrix (Fig. 6b). In this way, our 3D-CAE can generate three different kinds of porosity matrices. We choose the $3 \times 3 \times 3$ arrangement in this study to balance structural complexity and computational efficiency; nonetheless, our GAD-MALL can handle three different scaffold arrangements in principle.

For the labeled dataset, the labels (the elastic modulus ($E$) and yield strength ($Y$) of the corresponding scaffolds) are computed by the FEM, whose accuracy is verified through careful calibration with experimental data. The deviations between the experiment and simulation are confirmed to be less than 10% (Fig. 6c, d, e, f, g, h).

## 3D printing and compression tests

The performance of the powder has an extensive influence on the formation quality of the 3D-printed products. Spherical Ti6Al4V (Ti) powders with few satellite particles were observed (Supplementary Fig. 23), suggesting good flowability. The powder sizes of D10, D50, and D90 in statistics were 23.9, 37.8, and 58.5 μm, respectively. Ti scaffolds with a size of $6 \times 6 \times 6$ mm were additively manufactured by a laser powder bed fusion (LPBF) process using an EOS M290 machine (Supplementary Fig. 24) in this work. The processing chamber was filled with argon gas to avoid harmful reactions. The key LPBF parameters used were as follows: laser power of 280 W, laser scanning speed of 1200 mm s$^{-1}$, and layer thickness of 30 μm. After heat treatment at a temperature of 800 °C for 2 h and cooling in a furnace, the Ti scaffolds were surface treated by sandblasting. Ti6Al4V sand with an average grain size of 106 μm was used in the sandblasting process. The outer surface of the Ti scaffolds was uniformly blasted to remove the adhered powder particles, with a pressure of 0.6 MPa at the outlet of the spray gun. The relative density of the struts composing the Ti scaffolds was greater than 99.5% (Supplementary Fig. 25).

Supplementary Fig. 23 shows the pure zinc (Zn) powders; the powder sizes of D10, D50, and D90 in statistics were 10.2, 19.6, and 39.4 μm, respectively. Zn scaffolds of $6 \times 6 \times 6$ mm were processed using a BLT S210 machine (Supplementary Fig. 24). The processing chamber was filled with argon gas, and a gas circulation system was employed to inhibit the negative effect of vaporization during the LPBF process. The Zn scaffolds were fabricated with a laser power of 40 W, a laser scanning speed of 500 mm s$^{-1}$, and a layer thickness of 0.03 mm. Chemical etching with 5% nitric acid and 5% hydrochloric acid (room temperature, 2 min) was applied to remove the adhered powder particles, and the relative density of the struts composing the Zn scaffolds reached 98.5% (Supplementary Fig. 25).

Compression tests were conducted using an Instron machine (10 kN load cell) at a crosshead speed of 1 mm min$^{-1}$ at room temperature. The compression direction was parallel to the building direction. Three replicas were manufactured to ensure reproducibility.

## Numerical simulation parameters

We performed the compression simulation on a 32-core and 64-thread CPU (Intel Xeon Gold 6226R Processor) using ABAQUS/Explicit

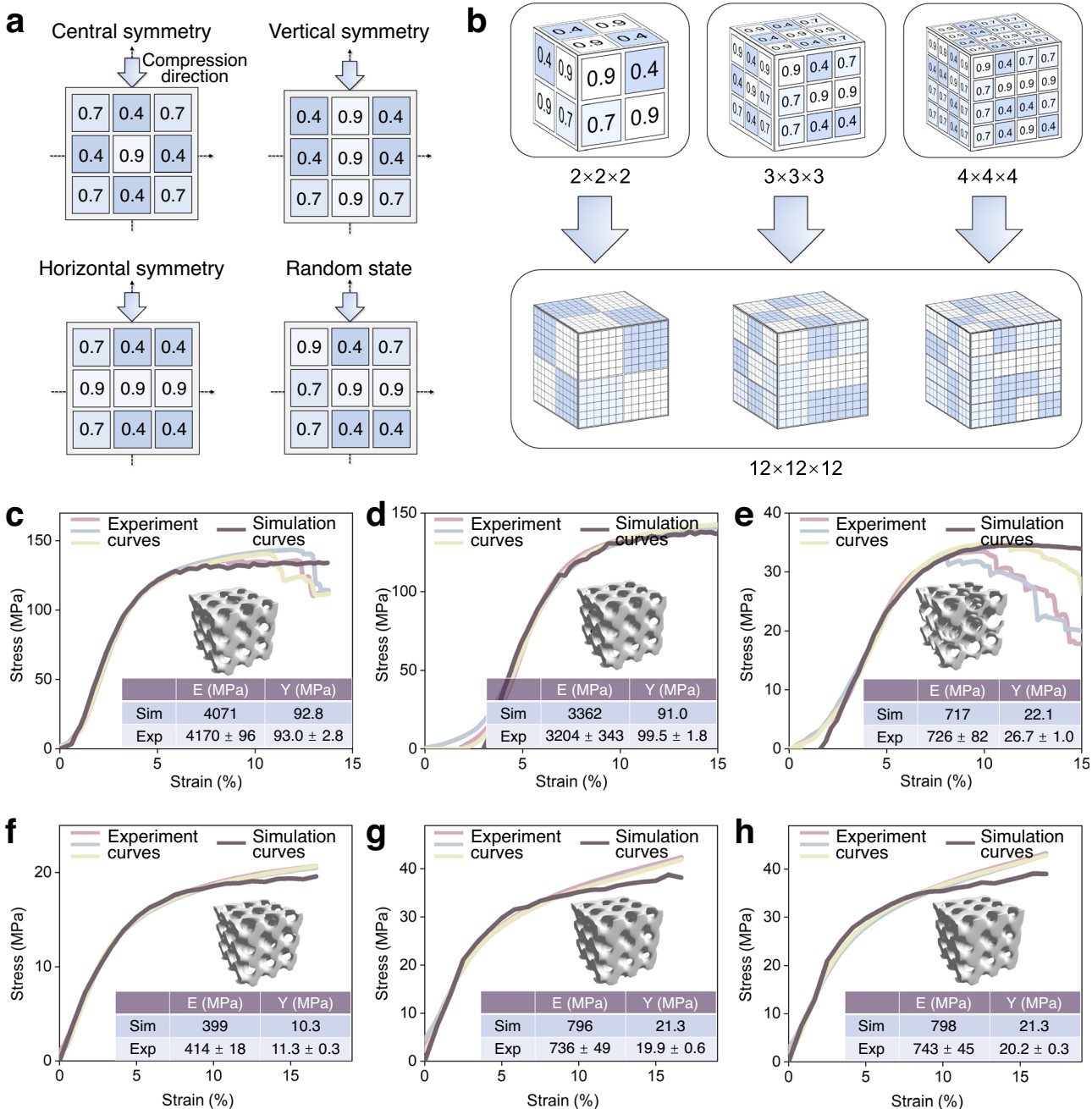

**Fig. 6 | Data generation and simulation calibration. a** The porosity matrix database contains three symmetries and one random arrangement. **b** Three kinds of porosity matrices: 2 × 2 × 2, 3 × 3 × 3 and 4 × 4 × 4, which can all expand to a 12 × 12 × 12 matrix. The 3 × 3 × 3 arrangement was chosen to balance structural complexity and computational efficiency. **a**, **b** show the schematics for unlabeled porosity matrix preparation. **c**, **d**, **e** The finite element method (FEM) simulation agrees with experimental observations. Three replicates were tested to ensure reproducibility. The error of the elastic modulus (*E*) and yield strength (*Y*) between the FEM simulation and experimental results is less than 10%. **c**, **d**, **e** refer to 3 Ti scaffolds with random shapes, and **f**, **g**, **h** refer to 3 Zn scaffolds with random shapes. All stress-strain curves are adjusted in the *x*-axis direction to make them overlap. Source data are provided as a Source Data file.

software[70]. The FEM was based on the same rigid-cylinder and deformable-implant-structure model. The material was homogeneous, and the Poisson's ratio was 0.25. The *E* was set to 5 GPa and the *Y* was set to 120 MPa based on the compression experiments of a pure Zn block prepared by LPBF (Supplementary Fig. 26). Ductile damage was used to simulate the plastic deformation to the failure stage. The fracture strain was set as 0.03, and the effects of triaxiality deviation and strain rate were neglected. We extracted displacements and forces in postprocessing and then converted them to strains and stresses, respectively.

## Machine-learning algorithms

The 3D-CAE consisted of an encoder and decoder. The encoder was composed of 3 3D convolutional layers (Conv3D). The input size was (12, 12, 12, 1). The first, second, third, and fourth layers contained 60, 30, and 15 filters, respectively. Three max-pooling layers between the convolutional layers were responsible for the downsampling. For example, one max-pooling layer reduced the size of Conv3D from (12, 12, 12) to (6, 6, 6), shrinking each (2, 2, 2) box to (1, 1, 1) and taking the maximum as its value. The size of the final layer was (3, 3, 3, 15). Another max pooling reduced it to the hidden representation

**Article** https://doi.org/10.1038/s41467-023-42415-y

(1, 1, 1, $x$), where $x$ represents the dimension. The decoder had the same Conv3D architecture, but with upsampling, the hidden feature (1, 1, 1, $x$) was converted back to (12, 12, 12, 1). The reconstruction loss was the mean square error (MSE) between the input and output.

The 3D-CNN model consisted of 3 convolutional layers (Fig. 5b). The first, second, and third layers contained 8, 4, and 2 filters, respectively; three max-pooling layers were located behind each convolutional layer. Finally, before reaching the output node, the last layer was flattened into 1048 neurons, followed by a series of fully connected layers (128, 64, 32). The activation function was the exponential linear unit. Moreover, the loss function was the mean square error. The program was written using Keras and TensorFlow[71]. We trained the 3D-CAE and 3D-CNNs using a GPU (NVIDIA GeForce RTX 3090 Ti) with 24 GB of memory.

## Reporting summary

Further information on research design is available in the Nature Portfolio Reporting Summary linked to this article.

## Data availability

The training datasets of the 3D-CAE and 3D-CNNs, and the FEM simulation dataset generated in this study have been deposited in the GitHub repository at https://github.com/Bop2000/GAD-MALL, ref. 72. Source data are provided with this paper.

## Code availability

The codes for the workflow of the GAD-MALL method, other state-of-the-art active learning algorithms, finite element methods and its automation pipeline, and the TPMS structure generation algorithm are publicly available in the GitHub repository at https://github.com/Bop2000/GAD-MALL, ref. 72.

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

## Acknowledgements

This work was funded by the National Key Research and Development Program of China (2018YFE0104200, P.W., Y.T., Y.Z.), National Natural Science Foundation of China (52175274, 82172065, and 51875310, P.W.), and Tsinghua-Toyota Joint Research Fund (P.W.) and Tsinghua Precision Medicine Foundation (P.W.); Y.W. would like to acknowledge the financial support of the Shuimu fellowship of Tsinghua University.

## Author contributions

Y.W. and Y.Q. conceived the idea; P.W. planned the study; Y.W. designed the machine-learning framework; Y.L. and B.P. wrote the relevant code; Y.Q. contributed to the digital design framework. J.D. and Y.Q. performed the FEM simulation and analysis; J.D., B.P., and A.L. performed the experiments; Y.W., B.P., J.D., and Y.Q. wrote the manuscript; B.P., Y.W., and Y.Q. designed and produced the figures. Y.T., L.H., and Y.Z. provided theoretical support. All authors participated in discussions and commented on the manuscript.

## Competing interests

The authors declare no competing interests.
