## [Peer Review File · Nature Communications]

REVIEWER COMMENTS

Reviewer #5 (Remarks to the Author):

1. The paper reports an active machine learning (AML) approach to understand the mechanism of the arrangement and enables the fast design to meet the specific performance requirements, which is kind similar to the paper by Li et al. (<https://arxiv.org/abs/2302.01078>). It means the framework is not completely new to the reviewer. Multiobjective Bayesian optimization (one kind of active learning approach) has been widely used, for example, in mechanical property optimization through formulation (Erps, et al 2021). The reviewer believes the above arxiv paper is closely relevant and should be cited.
2. The reviewer is not sure how are the 95 initial data points selected. Why 95?
3. The authors mentioned that the gyroid structures are generated in Matlab, and converted in to stl format, and then drew the mesh through Hypermesh, and input it to Abaqus in inp format. This process sounds not very automated. Can you integrate these separate modules to make the pipeline automated?
4. As the reviewer understands the paper wants to claim multi-objective design and optimization, however, Fig. 2C is more like a single-objective optimization. It might be better to show how multi-objective performance evolves with different iterations.
5. The insets in Fig. 3B are not very clear, especially the curve.
6. One question raised by reviewer #4 regarding ground truth is not completely addressed. As the FEM simulation couldn't guarantee the experimental measurement, it would be necessary to validate the so-called ground truth (FEM simulation). If the simulation and experimental measurement do not match with each other, then, the authors might have a risk of missing excellent candidates. In reviewer's opinion, using simulation as the ground truth should be very careful, especially when deal with 3D printing samples.

References

Erps, T., Foshey, M., Luković, M.K., Shou, W., Goetzke, H.H., Dietsch, H., Stoll, K., von Vacano, B. and Matusik, W., 2021. Accelerated discovery of 3D printing materials using data-driven multiobjective optimization. *Science Advances*, 7(42), p.eabf7435.

Response to the Reviewers' Report

We would like to express our sincere appreciation to the reviewers for the insightful suggestions and comments on our paper. Our response is structured in the following manner. The original comments from the reviewers or quotes from the original manuscript are copied below in black and italic font. Each comment is followed by a detailed response in blue font, and the corresponding manuscript modifications are indicated in red font. In the revised manuscript, the amended parts are highlighted in red font.

Reviewer #5:

1. The paper reports an active machine learning (AML) approach to understand the mechanism of the arrangement and enables the fast design to meet the specific performance requirements, which is kind similar to the paper by Li et al. (<https://arxiv.org/abs/2302.01078>). It means the framework is not completely new to the reviewer. Multiobjective Bayesian optimization (one kind of active learning approach) has been widely used, for example, in mechanical property optimization through formulation (Erps, et al 2021). The reviewer believes the above arxiv paper is closely relevant and should be cited.

Response:

We thank the reviewer for the comment. We wish to clarify that the paper by Li et al. was published in February 2023. However, the preprint of our study was, in fact, made available earlier on Research Square in 2022 (<https://doi.org/10.21203/rs.3.rs-2082876/v1>).

The mentioned references have now been included in the manuscript. Furthermore, we have conducted exhaustive Bayesian optimization on this task. The related setup and results are available in the supplementary material. Bayesian optimization did not yield favorable results, likely due to the **high dimensionality of the input**. The authors believe that the novelty of this manuscript is not about understanding the mechanism of the arrangement and enabling the fast design to meet the specific performance requirements but mainly lies in the following points: Our GAD-MALL method showed excellent performance at solving **high-dimensional multi-objective optimization problems**, which is a subject full of challenges and innovations. We also showed that GAD-MALL outperforms other current state-of-the-art approaches by a large margin.

Modification:

Two references have been added to the manuscript:

31. Li, B. *et al.* Computational Discovery of Microstructured Composites with Optimal Strength-Toughness Trade-Offs. *arXiv preprint arXiv:2302.01078* (2023).
32. Erps, T. *et al.* Accelerated discovery of 3D printing materials using data-driven multiobjective optimization. *Sci Adv* 7, eabf7435 (2021).

2. *The reviewer is not sure how are the 95 initial data points selected. Why 95?*

Response:

We appreciate the reviewer's comment. To clarify, the initial dataset comprised a random selection of 100 data points. However, five of these points encountered some errors during the mesh generation process in Hypermesh or during the simulation stage. As a result, the initial round effectively consisted of 95 data points, while the model already showed good performance (R^2 ratio ~ 0.9 and mean absolute error=3.1 on the test set).

Modification:

We added the following lines to the manuscript at **line 130**:

'since the predictive model based on this dataset already showed good performance on the testing dataset (R^2 ratio ~ 0.9).'

3. *The authors mentioned that the gyroid structures are generated in Matlab, and converted in to stl format, and then drew the mesh through Hypermesh, and input it to Abaqus in inp format. This process sounds not very automated. Can you integrate these separate modules to make the pipeline automated?*

Response:

We appreciate the reviewer's comment. Since our initial submission of the manuscript, we have pursued our goal of automating the pipeline and have successfully achieved this. The associated code can be accessed at <https://github.com/Bop2000/GAD-MALL>.

4. As the reviewer understands the paper wants to claim multi-objective design and optimization, however, Fig. 2C is more like a single-objective optimization. It might be better to show how multi-objective performance evolves with different iterations.

Response:

We thank the reviewer for the comment. The figure was now updated as suggested.

Fig. 2C. Baseline comparison of GAD-MALL with random search and Bayesian optimization. The red line represents the increase in yield strength, and the blue line represents the corresponding elastic modulus.

Modification:

Fig. 2C has been updated in the manuscript.

5. The insets in Fig. 3B are not very clear, especially the curve.

Response:

We thank the reviewer for the comment. We redrew Fig. 3B with improved resolution and quality.

Fig. 3B. The distribution of elastic modulus (E) and yield strength (Y) derived from the Finite Element Method (FEM) simulation results. The inset reflects the maximum yield strength value amongst the structures that fulfill the required elastic modulus criteria, drawn from the 20 selected structures in each iteration.

Modification:

Fig. 3B has been updated in the manuscript.

6. One question raised by reviewer #4 regarding ground truth is not completely addressed. As the FEM simulation couldn't guarantee the experimental measurement, it would be necessary to validate the so-called ground truth (FEM simulation). If the simulation and experimental measurement do not match with each other, then, the authors might have a risk of missing excellent candidates. In reviewer's opinion, using simulation as the ground truth should be very careful, especially when deal with 3D printing samples

Response:

We appreciate the reviewer's comments. We definitely agree with you and reviewer #4's remarks on potential problems of using FEM simulation as the database. Indeed, it should be very careful to use simulation results as the ground truth, especially regarding 3D printed samples, since the repeatability and credibility may be in question.

In response to these concerns, we have conducted comprehensive simulation and

experimental studies. These can be found in the revised manuscript (**lines 376 - 378** and **Extended Data Fig. 5 C-G**):

'For the labeled dataset, the labels (the elastic modulus (E) and yield strength (Y) of the corresponding scaffolds are computed by the FEM, whose accuracy is verified through careful calibration with experimental data. The deviations between the experiment and simulation are confirmed to be less than 10%.'

Extended Data Fig. 5 (C-H) simulation calibration. **Extended Data Fig. 5. (C to H) The simulation calibration (from the manuscript).** The FEM simulation agrees with experimental observations. Three replicates were tested to ensure reproducibility. The error of the E and Y between the FEM simulation and experimental results was less than 10%. (C to E) refer to 3 Ti scaffolds with random shapes, and (F to H) refer to 3 Zn scaffolds with random shapes. (C to H) All stress-strain curves are adjusted in the x-axis direction to make them overlap. ABAQUS/Explicit software was used for compression simulation.

And lines 607 - 608:

'Three replicates were printed and tested to ensure reproducibility. The error of the E and Y between the FEM simulation and experimental results was less than 10%. (Extended Data Fig. 5 C-G)'

Modification:

To make this important statement more visible to the audience, we added the following lines to the manuscript at **line 83**:

'An important prerequisite is that the simulated properties agree with experimental observation within an acceptable error range. Therefore, several replicates (to ensure reproducibility) of candidate materials were manufactured and tested, by which the experimental measurements (E and Y) were used to calibrate the FEM parameters such that the error between the simulation and experimental results was less than 10% (see extended data Fig. 5 C-G).'

REVIEWERS' COMMENTS

Reviewer #5 (Remarks to the Author):

The authors have carefully replied and addressed all the reviewers' concerns. It is believed that this manuscript can be considered for acceptance.

Response to the Reviewers' Report

Reviewer #5 (Remarks to the Author):

The authors have carefully replied and addressed all the reviewers' concerns. It is believed that this manuscript can be considered for acceptance.

Response:

We would like to cordially thank the reviewer for the valuable suggestions and comments on our paper.